# Degron Pathways and Leishmaniasis: Debating Potential Roles of *Leishmania* spp. Proteases Activity on Guiding Hosts Immune Response and Their Relevance to the Development of Vaccines

**DOI:** 10.3390/vaccines11061015

**Published:** 2023-05-23

**Authors:** Adriane Silva Oliveira, Lara Mata Aredes-Riguetti, Bernardo Acácio Santini Pereira, Carlos Roberto Alves, Franklin Souza-Silva

**Affiliations:** 1Laboratório de Biologia Molecular e Doenças Endêmicas, Instituto Oswaldo Cruz, Fundação Oswaldo Cruz, Manguinhos, Rio de Janeiro 21040-360, RJ, Brazil; 2Departamento de Patologia, Faculdade de Medicina, Universidade Federal Fluminense, Niterói 24033-900, RJ, Brazil; 3Centro de Desenvolvimento Tecnológico em Saúde, Fundação Oswaldo Cruz, Manguinhos, Rio de Janeiro 21040-360, RJ, Brazil; 4Faculdade de Ciências Biológicas e da Saúde, Universidade Iguaçu, Avenida Abílio Augusto Távora, 2134, Dom Rodrigo, Nova Iguaçu 26260-100, RJ, Brazil

**Keywords:** *Leishmania* spp., degron, cysteine proteases, serine proteases, immune response, cytokines, transcription factors

## Abstract

Degrons are short peptide sequences that signalize target sites for protein degradation by proteases. Herein, we bring forth the discussion on degrons present in proteins related to the immune system of *Mus musculus* that are potential targets for cysteine and serine proteases of *Leishmania* spp. and their possible roles on host immune regulation by parasites. The Merops database was used to identify protease substrates and proteases sequence motifs, while MAST/MEME Suite was applied to find degron motifs in murine cytokines (IFN-y, IL-4, IL-5, IL-13, IL-17) and transcription factors (NF-kappaB, STAT-1, AP-1, CREB, and BACH2). STRING tool was used to construct an interaction network for the immune factors and SWISS-MODEL server to generate three-dimensional models of proteins. In silico assays confirm the occurrence of degrons in the selected immune response factors. Further analyses were conducted only in those with resolved three-dimensional structures. The predicted interaction network of degron-containing *M. musculus* proteins shows the possibility that the specific activity of parasite proteases could interfere with the trend of Th1/Th2 immune responses. Data suggest that degrons may play a role in the immune responses in leishmaniases as targets for parasite proteases activity, directing the degradation of specific immune-related factors.

## 1. Introduction

Parasites of the genus *Leishmania* are protozoa transmitted to mammalian hosts by sandflies and are the causative agents of visceral (VL) and cutaneous (CL) leishmaniases, diseases widely distributed throughout the world [1]. There is currently no vaccine against this disease available, although vaccination of the population in endemic areas remains among the main strategies to pursue the prevention and control of leishmaniasis. Some vaccine prototypes are moving into clinical trials, but most remain in the early research stages.

Vaccine development for leishmaniases faces some difficulties since *Leishmania* spp. has a complex biological cycle, living in sandflies, humans, and many other animals. In fact, a vaccine only for humans would not eliminate these protozoans since they still circulate in other hosts. Furthermore, there is a challenge in interpreting data from animal models to apply to humans as the immune response to *Leishmania* spp. is multifaceted, and our current understanding of the influence of parasite virulence factors is limited. Therefore, exploring the molecular mechanisms of the parasite virulence factors, such as proteases, on host immunity could grant useful knowledge to surpass the challenges of vaccine development [2].

In fact, proteases of *Leishmania* spp. are major virulence factors in these parasites, and they are responsible for adaptive performance against hosts’ immune responses [3]. These enzymes act on the hydrolysis of peptide bonds for the degradation of proteins that participate in a series of biological processes [4]. By acting in networks of interactions between proteins/enzymes and activating or inactivating other enzymes, proteases may be involved in a synergistic complex of actions and reactions related to the host’s immune response [5,6]. In *Leishmania* spp., proteases provide a promising basis for such a system, exhibiting strong specificity for short cognate target sites, which can be recognized and cleaved, therefore participating in multiple physiological or virulence contexts [7].

These short cognates, whether conceptualized as degrons, fragments of proteins, or structural motifs, are important in regulating the rates of protein degradation [8] and, as such, may play crucial roles in modulating the immune system during parasitic infection by limiting immune factors availability. These sequences have been studied in multiple biological contexts, such as the synthetic circuit of proteins, where they are promising targets in processes such as targeted degradation [9], and they may also be relevant in physiological and virulence processes due to their ability to drive the degradation of specific proteins. In mammalian cells, targeted degradation may enable the suppression of specific proteins, as proposed in studies on the treatment of cancer patients [10].

Degrons were described as an important signal in the ubiquitin-proteasome system, which is the main route of selective protein degradation [11]. Ubiquitin is conjugated to the target protein and directs it to proteolytic degradation in the 26S proteasome, while the ubiquitin-ligase enzymes (E1, E2, E3) act by activating, conjugating, and recognizing the target substrate, respectively [12,13]. The role of degrons as selective markers in the proteolytic process of the 26S proteasome is poorly understood [14].

The functional relationship of *Leishmania* spp. proteases with immune responses are well-established, for example: (i) acting in the inhibition of interleukin (IL)-12 production by NF-kappaB cleavage [15], (ii) acting in the inhibition of production of nitric oxide by cleaving STAT-1 and AP-1 transcription factors [16], and (iii) acting in the degradation of major histocompatibility complex class II proteins [17]. However, recognition of host proteins by *Leishmania* proteases via degrons has not yet been properly mapped. Therefore, it is possible that degrons may direct parasite protease activity toward the modulation of the host’s immune response between Th1 and Th2 profiles. Once these interactions are mapped and understood, it may be possible better understand their effects on immune responses, thus contributing to the mapping of more specific vaccine targets.

To start comprehending the action of *Leishmania* spp. proteases directed by degrons, the present study applies a set of in silico detection and predictive analyses to propose the network actions of the proteases of these parasites on immune-related proteins of *Mus musculus*, which is the main experimental model for immune response studies in leishmaniases [18]. Data gathered here revealed degrons for *Leishmania* cysteine proteases (specifically, papain, cathepsin L, and cathepsin B) and serine proteases (specifically, Oligopetidase B and Prolyl oligopeptidase) occurring in cytokines (IFN-y, IL-4, IL-5, IL-13, IL-17) and transcription factors (NF-kappaB, STAT-1, AP-1, CREB, and BACH2) of *M. musculus*, indicating that the parasites enzymatic activity may influence Th1 and Th2 immune response profiles. The set of findings of this study opens a novel landscape for leishmaniases immune response driven by degron pathways.

## 2. Materials and Methods

### 2.1. Querying for Protease Sequence Motifs in Databases

The Merops database (Version 12.4—https://www.ebi.ac.uk/merops/ (accessed on 10 November 2022)) was used for retrieving protease cleavage site sequences for the following proteases: papain (C01.001), cathepsin L (C01.032), cathepsin B (C01.060), prolyl oligopeptidase (S09.001), and oligopeptidase B (S09.010). Queries were conducted with the following organisms: *Leishmania (Leishamnia) amazonensis* containing a total of 7 known and putative peptidases, and *L. (L.) major*, with a total of 101 known and putative peptidases. Proteases were selected accordingly to family, as (i) homologous to CPA peptidase (C1, papain, ID: MER0000647), (ii) homologous to CPB peptidase (C1, cathepsin L, ID: MER0002902), (iii) homologous to CPC peptidase (C1, cathepsin B, ID: MER0002901), oligopeptidase B (S9, ID: MER0000410), and (iv) homologous to prolyl oligopeptidase (S9, ID: MER0000392). Residues were selected from the specificity matrix and combinatorial peptides, as informed in the database. Subsequently, structural alignments of these selected proteases were performed using EMBOSS Water (Smith–Waterman algorithm—https://www.ebi.ac.uk/Tools/emboss/ (accessed on 10 November 2022)) to generate comparative models of their respective homologous proteases from *Leishmania* spp.

### 2.2. Searches for Degrons

MAST/MEME Suite tool (MEME Suits 5.5.0—https://meme-suite.org/meme/doc/mast.html?man_type=web (accessed on 22 February 2023)) was used to map degrons in the *M. musculus* sequences of cytokines (IFN-y: AAI19064.1; IL-4: AAH27514.1; IL5: NP_034688.1; IL13: NP_032381.1; IL-17:NP_849273.1) and transcription factors (NF-kappaB: NP_001170840.1; STAT-1: NP_064376.1; AP-1: NP_034721.1; CREB: NP_001004062.1; BACH2: NP_001103131.1) The settings were adjusted as follows: classic discovery mode; sequence alphabet (DNA, RNA, or protein), mode of distribution of motifs in the sequence (zero or one occurrence per sequence—zoops), and the number of motifs sought [3].

### 2.3. Interaction Network Prediction

To assess potential interactions between the groups of proteins that have been marked as containing degrons, interaction networks were generated using STRING (https://string-db.org/ (accessed on 05 January 2023)), with the following settings: network type—full-time STRING network; score required—0.400 mean confidence; and FDR stringency—high (1%). Interactions were organized using criteria-selected databases, experimental data, genetic neighborhood data, and coexpression data. The network was generated in 20 nodes, and, therefore, due to this scale, some functional partners of the immune response proteins previously selected were added to the final network structure.

### 2.4. Three-Dimensional Models

Linear sequences of cytokines and transcription factors were used to construct three-dimensional models. The models were generated using the SWISS-MODEL server (https://swissmodel.expasy.org/ (accessed on 15 January 2023)) based on the homology of proteins with resolved structures. Accepted identity values were between 40% and 100%. PyMOL server (https://pymol.org/2/ (accessed on 20 January 2023)) was used to view the generated models. 

## 3. Results

### 3.1. Degrons Mapping

Immune response-related factors were chosen for this analysis with the intention to encompass an array of pathways that can be crucial in the balance between Th1 and Th2 responses in *Leishmania* spp. infection. Therefore, we included in this study five cytokines (IFN-y, IL-4, IL-5, IL-13, and IL-17) and five transcription factors (NF-kappaB, STAT-1, AP-1, CREB, and BACH2), which were submitted to in silico analyses with the purpose of identifying target signaling motifs for proteases, the degrons (Table 1 and Table 2).

As the degron residues for papain, cathepsin L, cathepsin B, prolyl oligopeptidase, and oligopeptidase B proteases are known, and these enzymes correspond to proteases present in *Leishmania* spp., these motifs were queried in the immune response-related factors. 

The degrons found in the analyzed proteins are presented in Table 1 and Table 2, indicating the position of the amino acid residues in the target substrate (**n**—P4, P3, P2, P1, P1’, P2’, P3’, P4’—**c**). We observed in Merops a total of 15 degrons recognized by the analyzed proteases (4 for papain, 3 for cathepsin L, 4 for cathepsin B, 2 for prolyl oligopeptidase and 2 for oligopeptidase B). Furthermore, sequences matching the original database motifs, indicating potential degrons, could be observed in the sequences of the immune response-related factors: 15 matches for IFN-γ, IL-5, IL-13, IL-17, NF-kappaB, STAT-1, CREB, and BACH2; 10 matches for IL-4; and 7 matches for AP-1.

### 3.2. Degrons Location 

Cytokines and transcription factor sequences were used to construct three-dimensional models (Table 1). In general, the identity values obtained for the models were 58% or above: IL-5 (3b5k—94.7%), IL-13 (3lb6—58.2%), IL-17 (6hg4—68.9%), NF-kappaB (7cli—95.4%), STAT-1(1 × 1f—87.6%), AP-1 (5vpe—64.5%), CREB (5zko—66.2%), and BACH2 (3ohv–98.4%), with the exception of IFN-y (1fyh—50.88%) and IL-4 (2b8u—41.2%).

We could retrieve 138 potential degrons sequences from the linear sequences of the assessed murine proteins. However, as the selected three-dimensional prediction models do not cover the complete linear sequence of the proteins, fewer degrons could be effectively observed in the 3D predicted structures, with 39 degrons observed among the cytokines and 21 in the transcription factors (Figure 1).

In the cytokines IFN-y, IL-4, IL-5, and IL-13, degrons were located mainly in alpha-helix regions. Only in IL-17, the degrons, recognized by papain and oligopeptidase B, occur in beta sheet regions (Table 2). In addition, in IL-17, degrons recognized by prolyl oligopeptidase are present in the alpha-helix and coil regions.

In the transcription factor STAT-1, the degrons, recognized by cathepsin L and cathepsin B, are present in the beta sheet regions. It is interesting to note that multiple degrons can occur in each protein. In NF-kappaB, degrons are mainly present in the coil regions; in AP-1, only in the alpha helix; in CREB, degrons are present in the alpha helix; and in BACH2, they occur mainly in the beta sheet.

### 3.3. Proteins with Conserved Structures

The identity rates between the analyzed proteins of *Homo sapiens* (or other non-human primates) and those of *M. musculus* were assessed since the former was used as consulted sequences (Table 3). The results indicate high identity rates ranging from 83.1% to 98.2% between *H. sapiens* (and other non-human primates) NF-kappaB, SAT-1, AP-1, CREB, and BACH2 and their *M. musculus* homologs. Coverage was 100% in all analyses except BACH2 from *H. sapiens* (99%). As for IFN-γ, IL-4, IL-5, IL-13, and IL-17, the identity rates scored between 41% and 73.4%, and coverage values were between 75% and 100%. These results suggest that these immune response proteins are conserved in their structures and that the data obtained with the consulted sequences could reflect actual biochemical features of the *M. musculus* proteins.

### 3.4. Degrons Composition-Motifs

Motifs were selected by the lowest *p*-value (*p* < 0.05) for each target protein, as obtained using the MAST/MEME Suite tool. With this tool, it was possible to evaluate the probability of a better correspondence between the amino acids of each *M. musculus* protein and the respective degrons (Table 4). The lower values indicate a degron-specific signature on the transcription factor or cytokine sequence. Among the motifs observed, more specific signatures were found in transcription factors and in IL-5 and IL-17 cytokines. Similar amino acid motifs were found in IFN-y, showing glycine (G) residues for cathepsin B. IL-4 showed arginine (R), glutamic acid (E), isoleucine (I), and glycine (G) residues on the motif of OPB. IL-13 showed leucine (L) and lysine (K) for cathepsin B.

### 3.5. Network of Protease-Induced Interactions 

To illustrate how degron-driven degradation of host immune-related factors by *Leishmania* proteases could affect the overall immune response (with potential antagonic or synergic effects), we constructed a network of interactions among the *M. musculus* cytokines and transcription factors, with a scale of 20 nodes (Figure 2). These analyses confirm that cytokines and transcription factors interactions are specific (*p*-value: 1.47× 10^−9^) and that, among our study set, the cytokines with the higher numbers of connections in the network were IL-4 (13 connections), IFN-y (15 connections), and IL-13 (9 connections), while among the transcription factors, Jun (AP-1) showed the highest interaction in the network (12 connections). Patterns of interactions between Jun and IFN-y, as observed, suggest that this transcription factor is more related to a pro-inflammatory trend.

Additionally, functional partners, which emerged due to the criteria of the number of nodes analyzed in the network, can also point to directions where the cascade of effects potentially caused by protease activity may lead. Those include: the mitogen-activated protein kinases (Mapk9 and Mapk8), which are activated by pro-inflammatory cytokines, and act in cell proliferation, differentiation, migration, transformation, and programmed cell death; the nuclear phosphoprotein (Fos), which forms a complex with Jun and AP-1; the transcription factor Relb, which acts in complex with NF-kappaB; and the transcription factor ATF-2, which regulates the transcription of several genes, including those involved in anti-apoptosis, cell growth, and response to DNA damage.

## 4. Discussion

According to the available literature, degrons integrate a highly conserved system, present in a variety of organisms, which acts as directing protein degradation machinery. However, few degron motifs have been thoroughly identified [8,19]. Furthermore, even though degrons have been shown to contribute to the regulation of multiple physiological processes, such as immune system activity, studies on their contribution in the host–parasite interactions are very scarce [20]. With the aim to help fulfill such a gap of information, the present work showed the occurrence of degron motifs, both in murine cytokines and transcription factors that are specific targets for *Leishmania* cysteine or serine proteases and, therefore, may participate in driving the outcome of host immune responses during infections by these parasites. Such data bring forth novel insights that may be useful in selecting more promising vaccine targets.

Notwithstanding that degrons are mainly related to a protein quality control system in cells, which include degradation pathways to catalyze the refolding or removal of aberrant proteins [21], the exposition of these short peptides sequence onto the surface of immune system proteins can be used by *Leishmania* spp. as an escape strategy, to bend the host immune response profile toward a more favorable situation for parasite survival. 

*Leishmania* proteases have been described as participating in the biological cycles of these parasites by driving metabolic profiles and as escape mechanisms from the immune system of the mammalian hosts [22,23,24]. They are well-established pivotal virulence factors of these parasites [7,25], and their proteolytic actions are related to the composition of specific degradomes [25,26], including their respective substrate repertoire, which may include immune system-related proteins, as analyzed in this work. 

In fact, *Leishmania* spp. proteases, when secreted, may act as intracellular enzymes in the host cells [7], and it has been shown that these enzymes are actually released in small vesicles, nominated exosomes, to the *Leishmania* spp. extracellular environment. These proteases already have some known host–parasite interaction roles, such as promoting the modulation of host–parasite intercellular communication [27] and modulating cytokines expressed at the infection site [28,29].

Degrons are, in general, characterized as short continuous peptide sequences, and it was the standard definition applied throughout this study, but it is important to acknowledge that some may present non-continuous structures [30]. The continuous degrons sequences we have detected during our study present some distinctions concerning their amino acid sequences depending on the function of the protein they are located: residues of arginine, phenylalanine, glycine, and cysteine, were more frequent in transcription factors; while methionine, arginine, phenylalanine, leucine, and isoleucine were more frequent in cytokines. Lysine residues are usually located within degron segments to initiate protein degradation [31]; however, our findings suggest that other amino acids can participate in this process. 

In our analysis, degrons were found predominantly occur in alpha-helix regions of the studied immune response-related proteins. Furthermore, only in BACH2 and IL-13 were degrons observed in β-sheet and loop-like regions, respectively.

The mapping of degron motifs by our applied methodology indicates that both cytokines and transcription factors of *M. musculus* can have multiple degron motifs for cysteine or serine proteases within their sequences. Many functions carried out by interactions between immune system proteins can be regulated by these multiple degrons in target proteins [20,32]. Therefore, theoretical results provided here suggest that the degrons are present in important regions of these proteins and may direct to cleavage targets positions that can disrupt the function of the protein.

The function of these cytokines selected for our study in the immune response to leishmaniasis is well known, and their degron-driven degradation by parasite proteases may have considerable overall effects on the host responses against infection. In general, high levels of IFN-y are closely related to a protective response against infection, which can activate macrophages and nitric oxide production [33]. In addition, cytokines such as IL-17 can protect against *L. (L.) major*, as well as recruit neutrophils in target organs [34,35]. Conversely, the cytokines IL-4, IL-5, and IL-13 may contribute to susceptibility to *Leishmania* spp. infection. IL-4 and IL-5, for example, act in the manifestation of the Th2 response, causing the depletion of macrophages and accumulation of the intracellular parasite [36]. Thus, the catalytic effect on cytokines may affect the balance of Th1 and Th2 responses during infections by these parasites [37,38].

Additionally, the occurrence of degrons motifs associated with proteases in transcription factors that are also known to play important roles in the transcription of inflammatory mediators in *Leishmania* spp. infection is also an indicator of the potential impact they may have on host responses. Cleavage by proteases could inhibit signaling pathways related to those factors as a sophisticated mechanism to subvert the immune response [16]. In this context, the activation of NF-kappaB can direct it to the cell nucleus and influence a protective response against *Leishmania* sp. [39]. Furthermore, STAT-1-mediated signaling induces the production of IFN-y in macrophages, and when negatively regulated, it favors infection [40]. AP-1 regulates pro-inflammatory cytokines, chemokines, and nitric oxide production, and its inactivation is related to the activity of the leishmania GP63 protease [16]. In addition, BACH2 seems to play an intrinsic role in T lymphocytes, being related to the expansion and/or survival of T lymphocytes during infection by *Plasmodium chabaudi* and *Leishmania (Leishmania) donovani* [41].

The potential transcriptional regulation predicted here draws us to a growing perspective on host–pathogen interactions, pointing to possible mechanisms of epigenetic regulation of protein expression [42] via degron-mediated degradation. In addition, it is important to consider that although the in silico approach applied in this study generates a potential picture of the immunological interactions occurring in *Leishmania* spp. murine infection, this is strictly a predictive analysis and, therefore, requires further in vitro and in vivo experimental assays to confirm the data. The benefit of such analysis is to shed light on a new perspective on the complex immunological network occurring during these diseases’ development, which can be relevant for consideration in vaccine design.

## 5. Conclusions

Findings gathered herein provide innovative data on potent regulatory targets of immune response driven by *Leishmania* spp. proteases, as signaling motifs for the action of cysteine proteases and serine proteases of these parasites, occur in cytokines and transcription factors of *M. musculus*. The value of applying predictive approaches toward establishing a better understanding of the potential modulatory effects of *Leishmania* proteases on the immune response against these parasites, specifically focused on the occurrence of degrons motives influencing the degradation of proteins, is presented and discussed in this study.

The observation that multiple degrons for more than one proteinase class could be present in the same immune system-related protein suggests multiple possibilities of recognition for degradation during infection by *Leishmania* spp. Thus, attention to the active center of proteases that acts as amplifiers for Th2 response needs to be considered for developing an effective vaccine against leishmaniases. This hypothesis is reinforced since predictive analyses of the interaction network between the immune response proteins of *M. musculus* containing degrons suggest that specific actions of cysteine protease and serine proteases of the parasite may interfere with Th1 and Th2 infection profiles.

Collectively, the results presented here provide new insight into the immune response happening in leishmaniases, opening perspectives on the mechanism behind the host-pathogen interaction in different moments of infection based on degrons, revealing opportunities for using them as the target of more effective vaccines.

## Figures and Tables

**Figure 1 vaccines-11-01015-f001:**
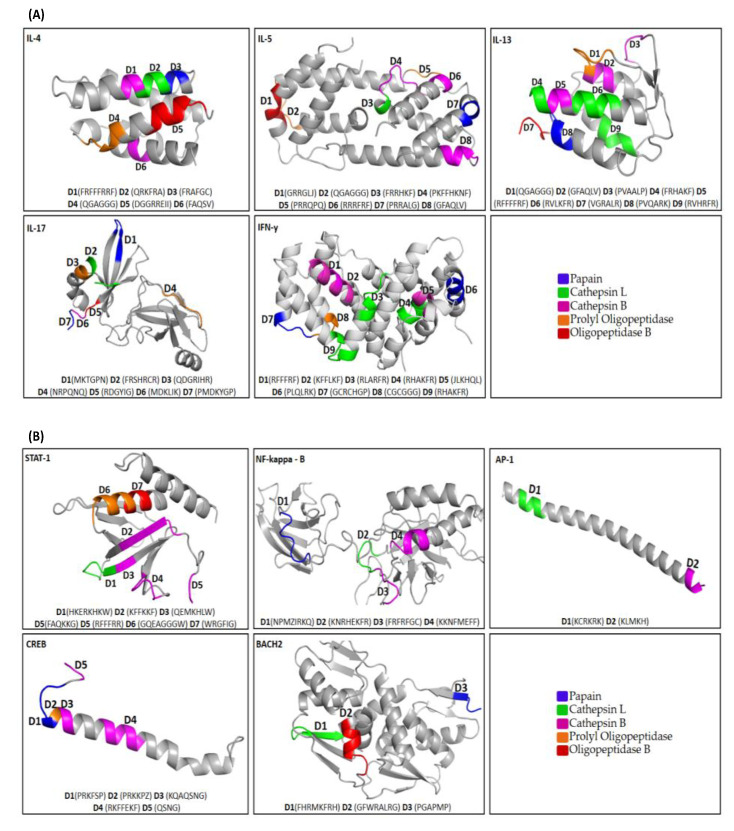
Location of degrons in the three-dimensional models of the target proteins. Degrons of cytokines (**A**) and transcription factors (**B**) were found as α-Helix, β-Sheet, and Coil regions of assessed models. The degron motif sequences (D1–D15) are shown in parentheses, signifying the specific composition for each target protein to a specific protease as papain (blue), cathepsin L (green), cathepsin B (Purple), prolyl oligopeptidase (orange), and oligopeptidase B (red). Sizes of the models are shown according to the settings of the PyMOL program. The proteins contemplate all regions of their sequence. Only in IL-17, a specific portion was selected due to its large size.

**Figure 2 vaccines-11-01015-f002:**
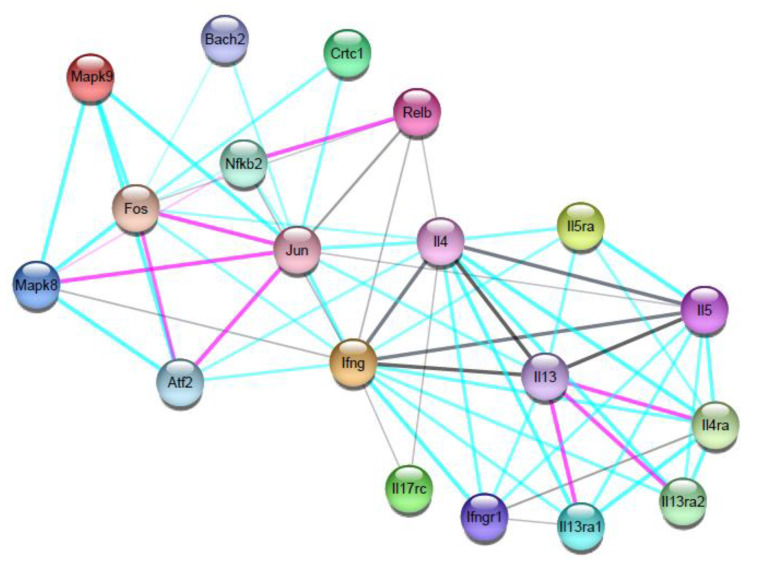
Network of interactions between proteins from *M. musculus*. The network was generated by cytokines and transcription factors with described roles in the immune response of *M. musculus* infected by *Leishmania* spp., and that have specific degrons for proteases. The network is made up of nodes and edges. Colored nodes represent first-layer proteins, and filled nodes represent proteins with known or predicted 3D structures. The thickness of the edges indicates a greater or lesser interaction between the proteins, and the colors indicate the type of evidence of interaction, as (**−**) selected data, (**−**) experimental data, (**−**) cooccurrence, and (**−**) coexpression data. The network proteins are described as: interferon gamma (Ifng); interleukin-4 (Il4); interleukin-5 (Il5); interleukin-13 (Il13); interleukin-17 receptor C (Il17rc); nuclear factor NF-kappa-B p100 subunit (Nfkb2); signal-transducing adapter family member 1 (Stap1); transcription factor AP-1 (Jun); CREB-regulated transcription coactivator 1(Crtc1); transcription regulator protein BACH2 (Bach2); and the functional partners: mitogen-activated protein kinases (Mapk9 and Mapk8); nuclear phosphoprotein (Fos); proto-oncogene, NF-KB subunit (Relb); cyclic AMP-dependent transcription factor (ATF-2).

**Table 1 vaccines-11-01015-t001:** Signaling degrons for cytokines hydrolysis.

Organisms	Proteases	Degrons																										
		Protease Amino Acid Residues Positions Described in Merops									Corresponding Amino Acid Residue Positions (*Mus musculus*)																	
											**IFN-y**									**IL-4**								
		P4	P3	P2	P1	P1’	P2’	P3’	P4’		P4	P3	P2	P1	P1’	P2’	P3’	P4’		P4	P3	P2	P1	P1’	P2’	P3’	P4’	
*Carica papaya*	Papain	P	P	V	R	A	S	G	P		-	G	C	R	C	H	G	P		-	-	F	R	A	F	G	C	
		P	P	V	Q	S	R	K	Q		-	P	L	Q	L	R	K	-		-	-	-	-	-	-	-	-	
		P	P	V	A	T	-	P	N		-	P	L	A	L	P	Q	-		-	-	-	-	-	-	-	-	
		P	P	V	A	T	R	P	N		-	P	L	A	L	R	P	-		-	-	-	-	-	-	-	-	
*Homo sapiens*	Cathepsin L	-	K	F	R	H	S	-	-		-	R	H	A	K	F	R	-		-	Q	R	K	F	R	A	-	
		-	R	F	R	S	H	-	-		-	-	R	L	A	R	F	R		-	F	R	A	F	R	C	-	
		H	K	F	R	H	-	-	-		-	-	F	R	A	H	K	F		-	-	Q	R	K	F	R	A	
*Homo sapiens*	Cathepsin B	-	-	F	R	F	F	-	-		-	R	F	F	F	R	F	-		F	R	F	F	F	R	R	F	
		-	-	Y	K	F	F	-	-		-	K	F	F	L	K	F	-		Q	K	F	F	R	K	F	-	
		H	L	M	K	-	-	-	-		-	J	L	K	H	Q	L	-		-	-	-	-	-	-	-	-	
		Q	S	V	G	F	A	-	-		-	C	G	F	A	Q	Q	L		-	F	A	Q	S	V	-	-	
*Sus scrofa*	Prolyl oligopeptidase	P	P	R	P	Q	P	Q	P		-	P	R	P	Q	P	Q	-		-	P	G	P	N	P	Q	-	
		G	Q	G	A	G	G	G	L		G	C	G	C	G	G	G	-		-	Q	G	A	G	G	G	-	
*E. coli*	Oligopeptidase B	D	G	R	R	G	Y	I	G		-	G	R	R	G	Q	J	-		D	G	G	R	R	E	I	I	
		G	V	G	R	S	S	R	G		-	I	G	R	L	Q	R	-		-	-	-	-	-	-	-	-	

Organisms	Proteases	Corresponding amino acid residue positions (*Mus musculus*)																										
		**IL-5**									**IL-13**									**IL-17**								
		P4	P3	P2	P1	P1’	P2’	P3’	P4’		P4	P3	P2	P1	P1’	P2’	P3’			P4	P3	P2	P1	P1’	P2’	P3’	P4’	
*Carica papaya*	Papain	-	P	R	R	A	L	G	-		-	P	V	R	A	L	G	-		-	P	M	D	K	Y	G	P	
		-	P	R	R	M	R	K	-		-	P	V	Q	A	R	K	-		-	P	M	D	K	R	K	Q	
		-	P	V	P	T	H	K	N		-	P	V	A	A	L	P	-		-	-	M	K	T	G	P	N	
		-	P	V	P	T	R	K	N		-	P	V	A	A	R	P	-		-	P	M	D	K	R	P	N	
*Homo sapiens*	Cathepsin L	-	R	H	S	K	F	R	-		-	R	V	L	K	F	R	-		-	H	D	K	F	R	H	-	
		-	M	R	R	F	R	L	-		-	R	V	H	R	F	R	-		-	F	R	S	H	R	C	R	
		-	F	R	R	H	K	F	-		-	F	R	H	A	K	F	-		-	F	R	V	H	W	F	-	
*Homo sapiens*	Cathepsin B	-	R	R	R	F	R	F	-		-	R	F	F	F	R	F	-		-	R	F	F	F	W	F	-	
		P	K	F	F	H	K	N	F		Q	K	F	R	H	G	F	F		P	K	D	F	Y	K	F	-	
		-	L	R	K	H	L	L	-		-	A	M	K	A	L	A	-		-	-	M	D	K	L	I	K	
		-	G	F	A	Q	L	V	-		-	G	F	A	Q	L	V	-		G	R	A	Q	G	V	G	F	
*Sus scrofa*	Prolyl oligopeptidase	-	P	R	R	Q	P	Q	-		-	P	R	P	Q	P	Q	-		-	N	R	P	Q	N	Q	-	
		-	Q	G	A	G	G	G	-		-	Q	G	A	G	G	G	-		-	Q	G	V	G	W	Q	-	
*E. coli*	Oligopeptidase B	-	G	R	R	G	L	J	-		-	G	R	R	G	L	I	-		-	-	R	D	G	Y	I	G	
		-	M	G	R	M	L	R	-		-	V	G	R	A	L	R	-		G	D	G	R	I	H	R	-	

**Table 2 vaccines-11-01015-t002:** Signaling degrons for transcription factors hydrolysis.

Organisms	Proteases	Degrons																										
		Protease Amino Acid Residues Positions Described in Merops									Corresponding Amino Acid Residue Positions (*Mus musculus*)																	
											**Nf-kappa-B**									**STAT-1**								
		P4	P3	P2	P1	P1’	P2’	P3’	P4’		P4	P3	P2	P1	P1’	P2’	P3’	P4’		P4	P3	P2	P1	P1’	P2’	P3’	P4’	
*Carica papaya*	Papain	P	P	V	R	A	S	G	P		-	P	G	R	D	G	G	-		-	P	K	R	A	P	G	-	
		P	P	V	Q	S	R	K	Q		N	P	M	Z	I	R	K	Q		-	P	V	Q	D	R	K	D	
		P	P	V	A	T	-	P	N		-	P	G	A	D	G	P	-		-	P	M	A	K	K	P	-	
		P	P	V	A	T	R	P	N			P	G	A	D	G	P	-		-	P	M	A	K	K	P	-	
*Homo sapiens*	Cathepsin L	-	K	F	R	H	S	-	-		K	N	R	H	E	K	F	R		-	F	R	H	S	K	D	-	
		-	R	F	R	S	H	-	-		F	R	G	F	R	F	R	Y		-	-	H	R	E	R	K	H	
		H	K	F	R	H	-	-	-		-	K	N	R	H	E	K	F		H	K	E	R	K	H	K	W	
*Homo sapiens*	Cathepsin B	-	-	F	R	F	F	-	-		-	F	R	F	R	F	G	C		-	R	F	F	F	R	R	-	
		-	-	Y	K	F	F	-	-		K	K	N	F	M	E	F	F		-	K	F	F	K	K	F	-	
		H	L	M	K	-	-	-	-		H	L	M	K	K	N	M	K		Q	E	M	K	H	L	W	-	
		Q	S	V	G	F	A	-	-		Q	M	E	G	F	I	Q	-		-	-	F	A	Q	K	K	G	
*Sus scrofa*	Prolyl oligopeptidase	P	P	R	P	Q	P	Q	P		P	P	E	P	Q	P	Q	-		-	P	K	P	Q	P	R	-	
		G	Q	G	A	G	G	G	L		-	P	G	A	G	G	G	-		G	Q	E	A	G	G	G	W	
*E. coli*	Oligopeptidase B	D	G	R	R	G	Y	I	G		-	G	G	R	G	G	I	-		-	-	W	R	G	F	I	G	
		G	V	G	R	S	S	R	G		-	P	G	R	D	G	R	-		P	G	R	A	P	R	-	-	

Organisms	Proteases	Corresponding amino acid residue positions (*Mus musculus*)																										
		**AP-1**									**CREB**									**BACH2**								
		P4	P3	P2	P1	P1’	P2’	P3’	P4’		P4	P3	P2	P1	P1’	P2’	P3’	P4’		P4	P3	P2	P1	P1’	P2’	P3’	P4’	
*Carica papaya*	Papain	-	-	-	-	-	-	-	-		-	P	V	R	A	N	G	-		-	P	G	R	P	M	G	-	
		-	-	-	-	-	-	-	-		-	P	V	Q	S	R	N	-		-	-	V	Q	D	R	G	Q	
		-	-	-	-	-	-	-	-		-	P	R	K	F	S	P	-		-	P	G	A	P	M	P	-	
											-	P	V	A	S	R	N	-		-	P	G	A	P	R	P	-	
*Homo sapiens*	Cathepsin L	K	C	R	K	R	K	-	-		F	R	H	M	E	N	R	I		F	H	R	M	K	F	R	H	
		-	-	-	-	-	-	-	-		-	D	R	V	H	R	E	R		-	F	R	A	H	R	F	-	
		-	K	C	R	K	R	N	-		-	-	F	R	H	H	K	N		F	R	H	H	K	F	R	H	
*Homo sapiens*	Cathepsin B	F	E	F	F	F	R	D	D		F	R	F	F	E	R	F	F		F	R	F	F	F	K	F	F	
		E	K	F	F	Y	D	D	-		R	K	F	F	E	K	F	-		F	K	F	F	I	K	F	F	
		K	L	M	K	H	L	N	-		K	Q	A	Q	S	N	G	-		F	Q	M	K	I	K	M	K	
		V	G	N	A	Q	N	V	G		-	F	A	Q	S	N	G			V	G	F	A	C	S	E	G	
*Sus scrofa*	Prolyl oligopeptidase	-	-	-	-	-	-	-	-		-	P	R	K	K	P	Z	-		-	P	G	P	P	P	Q	-	
		-	-	G	G	G	G	G	L		-	Q	G	A	G	G	G	-		-	P	G	A	G	G	G	-	
*E. coli*	Oligopeptidase B	D	E	R	R	G	Y	D	G		-	G	R	R	G	N	N	-		D	G	D	G	G	Y	N	-	
		-	-	-	-	-	-	-	-		-	M	G	R	S	N	R	-		G	F	W	R	A	L	R	G	

**Table 3 vaccines-11-01015-t003:** Alignments of immune-related proteins (cytokines and transcription factors) from mice, humans, and non-human primates.

Similarity Analysis								
*Mus musculus* Target Proteins	*Homo sapiens*				Primate (Not Human)			
	Coverage (%)	E-Value	Identity (%)	Access	Coverage (%)	E-Value	Identity (%)	Access
Nfkappa-B	100	0.0	96.2	NP_001070962.1	100	0.0	91.9	XP_004050061.1
SAT-1	100	9 × 10^−170^	83.1	NP_001304698.1	100	6 × 10^−176^	85.6	XP_045388373.1
AP-1	100	2 × 10^−178^	95.5	NP_002219.1	100	0.0	98.2	XP_003921548.1
CREB	100	0.0	86.3	NP_056136.2	100	0.0	86.0	XP_003796627.1
BACH2	99	0.0	90.1	NP_001164265.1	100	0.0	89.6	XP_037858189.1
IFN-Y	99	6 × 10^−36^	41.0	NP_000610.2	99	2 × 10^−45^	47.4	XP_012497489.1
IL-4	100	4 × 10^−23^	41.3	CAP72493.1	100	1 × 10^−26^	42.2	DP000644.1
IL-5	87	7 × 10^−58^	71.7	NP_000870.1	94	1 × 10^−62^	73.4	XP_012513070.1
IL-13	75	1 × 10^−30^	59.8	NP_002179.2	75	9 × 10^−34^	59.0	XP_012506775.1
IL-17	96	0.0	69.5	NP_703191.2	96	0.0	71.0	XP_012616045.1

**Table 4 vaccines-11-01015-t004:** Degrons matching motives in cytokines and transcription factors.

Target	Protease	Protease Motifs	Residues	Alignment of Motifs in the Targets	*p*-Value Alignment
Nf-kappaB	Cathepsin B	FRFRFGC	**7**	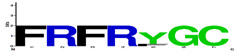	1.5 × 10^−9^
STAT-1	OPB	WRGFIG	**6**		1.5 × 10^−7^
AP-1	Cathepsin L	KCRKRK	**6**	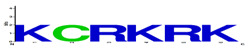	1.1 × 10^−5^
CREB	Cathepsin L	DRVHRER	**7**	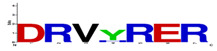	1.3 × 10^−8^
BACH2	Cathepsin L	FRAHRF	**6**	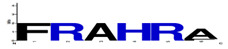	2.7 × 10^−8^
IFN-y	Cathepsin B	GFAQQL	**6**	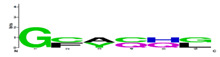	2.4 × 10^−7^
IL-4	OPB	DGRREIIG	**8**	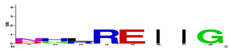	2.8 × 10^−7^
IL-5	OPB	MGRMLR	**6**	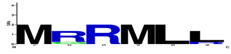	1.3 × 10^−7^
IL-13	Cathepsin B	AMKALA	**6**	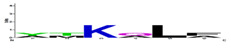	1.0 × 10^−8^
IL-17	Cathepsin L	FRSHRCR	**7**	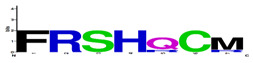	1.1 × 10^−9^

## Data Availability

Publicly available datasets were analyzed in this study. These data were accessed in web services repositories, as described in the material and methods.

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
