# Peer review of "Degron Pathways and Leishmaniasis: Debating Potential Roles of Leishmania spp. Proteases Activity on Guiding Hosts Immune Response and Their Relevance to the Development of Vaccines"

_vaccines, 2023, doi:10.3390/vaccines11061015_

Round 1
Reviewer 1 Report
This study provides innovative data suggesting that degrons may play a role in the immune responses in murine leishmaniasis, as targets for parasite proteases activity directing the degradation of specific immune-related factors. The study is well designed and organized, and the manuscript is well-written; however, the following minor points suggested are better to be considered.
Abstract
Line 35, replace spp by spp.
Introduction
Line 102, replace Mus musculus by M. muscuclus
112 Line, replace Leishmania amazonensis by Leishmania (Leishmania) amazonensis
Line 113, replace Leishmania major by L. (L.) major
Line 123, replace sp. by spp.
Line 127, replace Mus musculus by M. musculus
Results
Line 194, insert “humans,” before Homo sapiens (suggested)
Line 19d, insert “mice,” before Mus musculus (suggested)
Lines 197,199, replace Homo sapiens by H. sapiens
Line 209, replace Mus musculus by M. musculus
Discussion
Line 319, replace Mus musculus by M. musculus
Line 330, replace Leishmania major by L. (L.) major
Line 350, replace Leishmania donovani by L. (L.) donovani
Conclusion
Lines 358, 365, replace Mus musculus by M. musculus
Table 4, check thoroughly and arrange adequately
Figure 2 legend, replace Mus musculus by M. musculus (suggested)
Author Response
The reviewer#1 presented some very relevant and interesting suggestions, for which we are grateful. The suggestions were addressed throughout of reviewed version of the manuscript, as suggested by this reviewer as "tracking changes": “Line 35”, “Line 102”, “Line 112”, “Line 113”, “Line 123”, “Line 127”,”Line 194”, “Line 19”, ”Line 197”, “Line 199”, “Line 209”, “Line 319”, “Line 330”, “Line 350”, “Line 358”, “Line 365”, and “Figure 2”. Also, the authors clarify to this reviewer that "Table 4" is adequately arranged according to a correspondence between "log" and "degron sequence ". The degron sequence shown in the "logo graphic" column is an alignment based on the amino acids identity and represents the frequency of the aligned amino acids. Thus, only identical amino acids (2 bits) are more evident. For a better understanding of the readers, titles and column order have been changed ("Alignment of motifs in the targets", "Protease motifs" and "p-values motifs").
Reviewer 2 Report
1.- Write information about the interaction Degrons and Antibodies
2.-Why use only these cytokines for evaluating the Degrons´ response?
3.- Write information about NO and FNT alfa as response to Degrons.

Author Response
The reviewer#2 presented some very relevant and interesting questions and propositions, for which we are grateful, and would like to minutely address in our answers below:
Point 1: “1.- Write information about the interaction Degrons and Antibodies”
As a current definition, degrons are described as short peptide sequences that signalize target sites for protein degradation by proteases. Therefore, these short peptide sequences interact directly with the proteases and, from an immunological perspective, would not require the mediation of a bound-antibody to signalize the degradation site for the proteases. Hence, we have not explored any potential effects of antibodies in the context of our study. However, we consider that the suggestion of the reviewer regarding degrons and antibodies is interesting, as one could hypothesize that it is possible that degrons occur in conserved regions of the immunoglobulin sequences and, thus, may be another node in the complex web of interactions among host’s immune factors and parasite’s proteases, although such possibility has not been yet adequately addressed and, presently, would divert from the scope of this manuscript.
Point 2: “2.-Why use only these cytokines for evaluating the Degrons´ response?”
The reviewer is absolutely correct to inquire about analyzing other cytokines, as immune responses are highly complex and many factors contribute to the final outcome. However, we have opted to analyze this specific set of cytokines, as these cytokines are known to play relevant roles during mice infection by Leishmania spp. and there are multiple references in the literature on this regard to subsidize their selection. As the present manuscript is focused primarily on proposing a role for degrons in the interactions between Leishmania spp. and murine hosts, we have made a choice to prioritize, in this first study, the analysis of a more strict group of cytokines, but with well-established roles for leishmaniasis. Nevertheless, we completely agree with the reviewer's remark that analyzing the occurrence of degrons in more cytokines could lead to very important data and enrich significantly the knowledge on parasite-host interaction and intend to expand our study to include increasing numbers of cytokines in future developments of our research.
Point 3: “3.- Write information about NO and FNT alfa as response to Degrons.”
In regards of the reviewer’s request on analyzing degrons in TNF-alpha, we agree that this analysis can lead to interesting data, but as stated above, we considered that in this first presentation of the hypothesis of degrons affecting host-parasite interactions, a more limited number of selected cytokines but with well-defined roles in immune responses to leishmaniasis, would be a more robust approach. And TNF-alpha is a natural candidate for inclusion in our further studies on this matter.
As for NO, as it is a very simple gas molecule, it can not bear any degrons and, thus, could not adequately fit in our study. But, the Nitric Oxide Synthases (NOS), both the constitutive and the inducible ones, are complex proteins with important roles in immune responses, and, therefore, are suitable candidates to be included in further studies for degrons presence when our research group expands these pioneer ideas proposed in the present manuscript. Furthermore, it is important to comment that the purpose of this work is a prediction of degrons in proteins related to the immune response and thus, we apply an in silico strategy to identify these regions. However, relating the influence of cytokines and transcription factors on NO synthesis would be better observed in vitro assays, which is not the purpose of this work.